

# Mesoscale processes regulating the upper layer dynamics of Andaman waters during winter monsoon

*Salini Thaliyakkattil Chandran[1], Smitha Bal Raj[2], Sajeev Ravindran[1], Midhunshah Hussain[1], Muhammed Rafeeq[3]

1 Cochin University of Science and Technology, Kochi, 682016, India

2 Centre for Marine Living Resources & Ecology, Kochi, 682037, India

3 Center for Environment & Water, Research Institute, King Fahd University of Petroleum & Minerals, Dhahran 31261, Saudi Arabia

*Corresponding Author. email: salinitc@gmail.com, ph.+91 984688249

## 1  Abstract

The characteristic of cold core eddies and its influence on the hydrodynamics and biological production in Andaman waters were studied using in situ and satellite observations. The specific structure and patterns of the temperature-salinity (T-S) profiles, nutrients and chl a indicate the occurrence of the eddy, the spatial extent of which is well marked in sea surface height anomaly (SSHA). The Cyclonic Eddies are centered at 7°N and 86°E, 13°N and 88°E and 13°N and 93°E (CE1, CE2 and CE3 respectively). In situ measurements are done in the eastern flank CE1 along 8°N and 92.5-93.5°E. Vertical currents recorded using Acoustic Doppler Current Profiles (ADCP) shows northward flow along the track ($0.3$ m s$^{-1}$) while along the western flank, the flow is weak and southward. This evidence the occurrence of cyclonic eddy and the altimetry derived SSHA depicts the spatial extent. Analysis to explore the possible forcing to induce the occurrence of eddy, indicate baroclinic instability (Ri <0.0001) in the water column due to vertical shear in the horizontal flow. Bay of Bengal (BoB) water evidenced from the T-S profiles and the semi-annual Rossby wave are the contributing factors of eddy formation. Whereas, the wind stress curl is not a major inductive of divergence in the region. The eddy influenced the nutrient pattern ($NO_2$, $NO_3$, $PO_4$ and $SiO_4$) and the biological production (chl a) in the region though the influence is less significant. CE1 and CE2 are similar in terms of forcing mechanisms while, CE3 is associated with convective mixing processes occurring along the northwest coast of Andaman due to the prevalent cold dry continental air from north east.

## 20  Introduction



The Sea around the Andaman and Nicobar Island chain is influenced by reversing monsoon with
moisture rich summer winds and dry continental air flow from north east during winter (Potemra
et al., 1991).  The region receives enormous runoff and suspended matter from Ayeyarwady –
Salween river system, which has got significant influence on the hydro-dynamics and
oceanography (Robinson et al., 2007).  The region is characterised with strong stratification,
preventing vertical mixing causing lack of availability of nutrients in the upper layers resulting
oligotrophy.  The seasonal winds, moderate or strong, though experience during the summer and
winter months, are not found to exert any divergence or positive curl and nutrient pumping to
enrich the biological production is least encountered for the waters.  The sea is less productive
compared to Arabian sea and Bay of Bengal and the average primary production during fall
inter-monsoon is 283.19 mg C $m^{-2}$ $day^{-1}$ followed by spring inter-monsoon (249 mg C $m^{-2}$ $day^{-1}$),
summermonsoon (238.98 mg C $m^{-2}$ $day^{-1}$) and wintermonsoon (195.47 mg C $m^{-2}$ $day^{-1}$)
[Sanjeevan et al., 2012]. Earlier observations show that the eastern and western part of the island
chain is governed by distinct water properties, when west shows the typical BoB (Bay of Bengal)
characteristics, Northeast is highly influenced by the Ayeyarwady and Salween river system and
the southeast by the productive environment of Malacca strait (Salini et al., 2010). The region is
least explored for the oceanic processes, and the surveys conducted so far for understanding the
biodiversity and the basin scale environment associated with the living resources indicate,
absence of any major or seasonal processes, that results in nutrient pumping to alter the
production pattern. However, with the emergence of satellite techniques, especially the Altimtery
and ocean color imageries information on mesoscale to basin scale that contribute to the
understanding of the upper layer dynamics have been strengthened. Explanations have come on
such major processes in the Bay of Bengal, especially on number of eddies and gyres and also
the impact of cyclones which causes enormous mixing in its path. Eddies are mesoscale
processes (50-200km diameter), and ubiquitous feature of the ocean occurs in both clock wise
and anti-clock wise direction resulting convergence/divergence at the centre.
Mesoscale eddies play a dominant role to transport salt, heat and nutrients within the ocean
(Dong et al., 2014) and enhances the local production in generally oligotrophic areas (Hyrenbach
et al. 2006) ultimately influencing the production pattern in each trophic level (Bakun 2006).
Mechanism behind the eddy formation is suggested by many researchers. Different driving
mechanism have been attributed for the eddy formation such as Ekman pumping, remote forcing
from the equatorial Kelvin wave reflecting off the eastern boundary as Rossby wave.  According





to Yu et al. (1999), westward propagating Rossby wave excited by the remotely forced Kelvin
wave contribute substantially to the variability of the local circulation in ocean. Using the
multilayer model, Potemra et al. (1991) described coastal Kelvin wave, which originates at the
equator, propagating around the entire western perimeter of the region around both the Andaman
Sea and the Bay of Bengal. Mesoscale eddies are observed in the coastal waters of the Andaman
and Nicobar Islands (Hacker et al. 1998; Chen et al. 2013) based on in situ hydrographic
measurements. Burnaprathepart (2010) described the presence of eddy in Andaman Sea and its
role in enhancing the primary productivity synthesizing number of vertical profiles on chl a,
major nutrients, temperature and salinity. The eddy is identified based on the SSHA imagery and
the geostrophic current pattern indicating the low and the anticlockwise circulation pattern
resulting divergence and upsloping in the center.  The present study, based on a suit of in situ and
satellite on physical, chemical and biological measurements, explains the characteristics,
generation mechanism and evolution of the eddy and its impact on the regional primary
production.
**Data and Methodology**
In situ measurements were taken during FORV Sagar Sampada cruise 292 of 21Nov-14 Dec
2011. The environmental characteristics are understood from the station based measurements in
the east and west of the island chain.  However, focus is given for a transect with 4 stations (Fig.
1) along the eddy periphery, which was observed to be a detached feature from a major eddy
centered at 7°N 90°E. The meteorological parameters like air temperature, air pressure and
humidity were also collected through the instruments/sensors attached to the IRAWS onboard in
15 minute interval.  Profiles of temperature, salinity, dissolved oxygen and Sigma-t were
obtained using SeaBird 911 Plus CTD with Niskin water samplers and deck unit for data
acquisition. The datasets are processed for 1m bins.  Salinity is also derived from water samples
collected through Niskin samplers and using Guildline 8400A Autosal Salinometer to validate
the CTD derived data. Twelve numbers of 10 liter Niskin water samplers were used to collect
water samples from standard depths (surface, 10m, 20m, 30m, 50m, 75m, 100m, 120m, 150m,
200m 300m, 500m, 750m and 1000m) for the measurements of dissolved oxygen and nutrients.
Temperature-Salinity profiles for the watermass characteristics are based on averaged
(climatological) data from Levitus et al. (1994). Monthly composite of the chlorophyll data is
obtained from the Distributed Active Archive Center (DAAC) of National Aeronautics and



Space Administration, NASA. Dissolved oxygen was measured by Winkler titration. The
analysis of nitrite, nitrate, phosphate and silicate were done using a Skalar Analyser.
The bathymetry of the region is analysed using the NIO's modified dataset (Sindhu et al, 2007).
The abyssal plain with an even floor is located in the region. NIO modified the original ETOPO5
and ETOPO2v2 bathymetric grids in shallow water regions using the digitized data.
Wind stress curl (daily) is taken from ASCAT processed by NOAA/NESDIS utilizing
measurements from the scatterometer instrument aboard the EUMETSAT Metop satellites with a
spatial resolution of 25km. Chl a is taken from MODIS Aqua Level 3 at a spatial resolution of
4km  which is downloaded from Ocean Color Website and processed using SeaDas. SST is taken
from MODIS Aqua Level 3 at a spatial resolution of 4km which is downloaded from Ocean
Color Website. SSHA data is obtained with 7day temporal resolution from AVISO for the period
from January 2003 to January 2013. The cold core eddy is recognized through SSHA with
geostrophic current imagery obtained from https://oceanwatch.pifsc.noaa.gov and is centered at
7°N and 90°E with current moving in cyclonic direction. Net heat flux, Solar radiation, latent
heat flux and specific humidity are obtained from http://oaflux.whoi.edu.The in situ observation at
8°N along 92.5°E-93.5°E, is identified as the eastern periphery of the eddy identified.
Vertical sections of currents are derived using hull mounted OS II BB ADCP of 76.8 KHz
frequency operated along the ship's track.  Current datasets are acquired using VmDas in 8 m
bins and ensemble time of two seconds.  The ship heading and navigational informations are also
recorded while acquiring the raw data.  The first bin record of current started at 16m depth.  The
data in earth co-ordinates were postprocessed using WinADCP, for an ensemble period of 1
minute. Processed data which have a percentage good more than 80% only are considered for the
analysis.
Wavelet transform is an appropriate analysis tool to study multi-scale, non-stationary processes
occurring over finite spatial and temporal domain. Here the wavelet used to analyse time series
data of oceanographic parameters that contain non-stationary power at many different
frequencies. This technique is used to decompose the time series into its frequency components
based on the convolution of the original time series with set of wavelet functions and possible to
determine both the dominant modes of variability and how those modes vary in time. It expands
functions in terms of wavelets, which are generated in the form of translations and dilations of a



fixed function called mother wavelet. Meyers et al (1993) used wavelet analysis to study the
propagation of mixed Rossby-gravity waves in an idealized numerical model of the Indian
Ocean.
The phase speed for long baroclinic Rossby wave is given by $C = \frac{-gH_0\beta}{f^2}$          (1)
where g is the reduced gravity term (taken as 0.04 m s$^{-2}$ for the first baroclinic mode), $H_0$ is the
thermocline depth (taken as an annual mean depth of 20$^\circ$C isotherm derived from Levitus et al.,
1994), f is the Coriolis parameter, and $\beta = \frac{\partial f}{\partial \phi}$ , $\phi$ is the latitude

## Results and Discussion

### Physical characteristics of the eddy region (Eddy dynamics)

The region is characterized with warm (27.6-28°C), humid (72-77%) air and wind is from
northeast suggesting the prevalence of northeast monsoon condition of magnitude between 10-
12m s-1  with comparatively lower speed (10m s-1 ) in the western part  and higher speed in the
eastern part of the eddy (which is named thereafter as CE1).
The SST varies between 28.4-28.8°C with lower temperature near coastal water comparing to
offshore. The surface salinity (33.00) and density (20.40 kg m$^{-3}$ values are same in coastal and
offshore waters. The regional watermass characteristics from temperature, salinity and density
profiles show that the area is occupied by Bay of Bengal (BoB) waters with temperature 28.0-
28.5°C, salinity 33.2-33.8 and density 20.6-20.8 kg m$^{-3}$. Vertical temperature distribution along
8°N shows warm (>28.5°C) and thick isothermal layer (~54m) in the western part and it
showed a gradual decrease towards east (20m) (Fig.2b). The most important feature in the
thermal structure is the upsloping of isothermal layer and is prominent in the subsurface (54-
220m) also and the mixed layer depth (MLD) shoaled from west to east (47-19m). The vertical
salinity and density distribution show the presence of low saline (32.9-33.1) in the upper
30m, with an upsloping tendency (Fig.2c, d) as in the case of temperature. Similar pattern is also
reflected in density characteristics too.
The vertical current structure at 8°N along 92.5°E to 93.5°E (Fig.3) shows irregular current
pattern from surface to 90m.  Along the eastern part of the 100km transect, major flow is towards



south (≅30km), west to it with a narrow and weak northward flow, followed by major southward
drift up to 40m.  However, the response to this irregular pattern is insignificant in the T-S
profiles, and so the eastern part of the transect (~60km) is not considered for addressing the
eddy.  In the western flank, the northward and the subsequent flow towards indicate the cyclonic
flow direction.  The current recorded at 16m depth is considered for near surface pattern and this
shows the presence of a northern component with magnitude 0.3m $s^{-1}$ in the eastern part and
negligible speed in the western part and it directed towards west. But at 40m the current
magnitude is decreasing in the eastern flank (0.1 m $s^{-1}$) and increasing magnitude in the western
flank (0.1 m $s^{-1}$) with direction changing from northeast to southwest. The current at 88m also
follow the same pattern but magnitude changes from 0.5 m $s^{-1}$ in the western part and 0.4 m $s^{-1}$ in
the eastern part. The upsloping in the T-S profiles concurrent to this confirms the feature as a
subsurface cyclonic eddy.  The flow in the eastern flank is towards north (0.3 m $s^{-1}$) and at west
it is to south (0.5 m $s^{-1}$). The data is analyzed for all 8m cells up to 88m depth, and found to
follow the same pattern as that of near surface but with a decreasing magnitude. Below 88m the
dataset contains spurious values and so discarded.
**Generation Eddy Mechanism**
The possible physical mechanism that govern the eddy includes the wind stress curl, topographic
instability, shear flows, baroclinic instability and the radiation of Rossby waves from pole ward
propagating coastal Kelvin waves etc. (White, 1977; Kessler, 1990).  Daily wind stress curl is
examined to identify the local forcing that contributes to the formation and sustenance of the
eddy. Curl of the eddy region from ASCAT wind data shows negative values in the range
between-5.6x $10^{-8}$ to -8.24x$10^{-8}$ Pa $m^{-1}$, indicating convergence and hence the contribution due to
wind stress curl is ruled out.
Other possibility of eddy generation mechanism is the differential mixing of region with the
adjacent sea mainly through inflow from Malacca Strait and freshwater influx from adjoining
rivers leads to strong density variations in the water column. This variation may reduce or
enhance the mechanical effects in the form of eddy or meanders in the region.  This is measured
based on the estimated Richardson Number (Ri). According to Miles (1961) the flow is stable if
Ri>0.25.





Ri is calculated as $Ri = \dfrac{N^2}{(\frac{\partial u}{\partial z})^2}$                    (2)
where $N^2$ is the Brunt Vaisala frequency (BV)
$N^2 = \dfrac{-g}{\rho_0}\dfrac{\partial \sigma_t}{\partial z},$                    (3)
g is the gravitational acceleration, $\rho_0$ is the average sea water density, z is the depth,$\sigma_t$ is $\rho$ -1000
where $\rho$ is the sea water density. The denominator term $\partial u/\partial z$ is velocity gradient which is an
indicator of strength of mechanical generation calculated from vertical current profiles acquired
using ADCP.
The low BV (avg $3.165 \times 10^{-5}$ s$^{-1}$) and large velocity gradient (avg 3.968 s$^{-2}$) resulted into low Ri
(avg 0.0001) indicating unstable well mixed water column. These leads to instability in the water
column and favors eddy like perturbation in the region.
Instability arises either as a result of mixing of different water masses or due to the shear flows.
Mixing with other water masses can be ruled out as we have a clear evidence of presence of BoB
water in the eddy region from the T-S profiles.  Other option is the prevalence of any planetary
waves that modulate the horizontal flow and to induce shear and thereby instability. And such
instability has been well reported along this region by Schott et al., 2009 and Rao et al., 2010that
planetary waves influence the near surface circulation through local and remote forcing. The role
of this planetary wave influence on the eddy generation mechanism is examined using altimeter
data and mapping the planetary wave propagation to identify their influence on regional
circulation.  Referring Yu (2003), the Hovmuller diagram of SSHA at 8°N along 89°E to 94°E is
analyzed to track the planetary wave are plotted (Fig.4). Low SSHA in this region from mid-
November to mid-January indicates the presence of upwelling mode Rossby wave
(Gireeshkumar et al., 2011). Negative SSHA is almost horizontal indicating a fast propagation of
Rossby wave. Further west (near to the eddy location) negative SSHA showed a steeper slope,
indicating a slower propagation. The westward propagating signal took about 45-60 days to
travel from the coast of Nicobar Island chain (Potemra et al., 1991) to the core of the eddy
region, which yields the phase velocity of the westward signal at 0.20 m s$^{-1}$ . The theoretical
phase speed of Rossby wave at 8°N which propagate westwards is calculated as 0.21 m s$^{-1}$ . This
suggests that the signal appearing in the plot is a Rossby wave that has been generated on the



west coast of Nicobar island chain. The estimated speed of the wave is close to the theoretical
wave speed and the estimate also compares well with earlier results of Yang et al. (1998),Yu
(2003), Rao et al. (2002) and Gireesh Kumar et al. (2011). The Rossby waves were produced by
radiation from the west coast of Nicobar Island chain in association with poleward propagating
coastal Kelvin waves (Potemra et al. 1991,). Nuncio and Prasanna Kumar (2012) suggested that
the interaction of westward propagating Rossby waves and local wind stress curl cause
baroclinic instability and meandering in Bay of Bengal to induce eddy like features. Using a
numerical model, Kurien et al. (2010) also concluded that baroclinic instability plays a key role
in meander growth and eddy generation in BoB. Srinivas, et al (2012) argued that coastal Kelvin
waves and the associated radiated Rossby waves from the east play a dominant role in the
mesoscale eddy generation in Bay of Bengal.
To ascertain the periodicity of SSHA, the data is again subjected to continuous wavelet
transforms with Morlet wave as mother wavelet following Torrence and Compo (1998). It is
understood from the Fig.6 that the dominant mode of variability is semiannual. In the Andaman
waters the wave period is more variable due to the effect of westward propagating Rossby wave
from the coastally trapped Kelvin wave (Vialard et al., 2009 and Nienhaus et al., 2012).  From
the power and global wavelet spectrum (Fig.5), the predominant frequencies are at semiannual
and annual modes. The annual mode seems to be reduced in intensity compared to the
semiannual mode. On the basis of the results of wavelet analysis, wecould state that the
semiannual Rossby waves are significant in the years 2005, 2008, 2010 and 2011, where the
annual wavelets are significant in 2006-2009.Therefore, we concluded that the westward
propagating Rossby wave radiated from the coastal Kelvin wave contribute to cyclonic eddy in
the region.
**Chemical and biological response of the eddy**
Concurrent with the thermohaline oscillations, the vertical structure of dissolved oxygen (DO)
also demonstrate fluctuations above 90 m depth. The 4.22 ml/L DO contour shoaled from a
depth of about 47 m (92.3°E) to 25–30 m at eastern flank of the eddy (93.3°E).  The upper nitrate
($NO_3$) concentration is in the dectectable levels (0.67 µM-0.98 µM) and shows slight upsloping
towards the eastern flank (93.3°E).  The phosphate ($PO_4$) concentration in the upper water was
also at a detectable level and showed a slight upsloping towards the eastern side (0.12 µM at
92.3°E and 0.27 µM at 93.3°E). The vertical distribution of silicate ($SiO_4$) also showed slight





upsloping towards the eastern periphery (0.77 μM at 92.3°E to 1.62 μM at 93.3°E) [Table. 1].
Hence, concomitant with the thermohaline characteristics, the vertical distribution of nutrients
also showed oscillations in the upper water column.
The physical and chemical characteristics do reflect on the regional biology and this is well
reflected in the surface chl a distribution. The chl a derived from ocean colour imagery (Fig. 6)
can be illustrate the standing stock of the primary consumers for the optical depth and is 0.5 mg
m$^{-3}$ in the eddy region compared to the nearby region (0.1 mg m$^{-3}$). This increase within the eddy
in association with the nutrient values, explains the impact of churning due to the eddy. And this
point out the relevance in occurrence of such mesoscale processes to influence the production
marginally in the Andaman waters.
**Satellite evidence (SSHA based) for cyclonic eddies**
General purview on distribution of such mesoscale production favourable pockets is examined
using monthly SSHA pattern (Fig.7a-d) for the winter monsoon (Nov-Feb) of 2011.   This
evidenced the presence of three cyclonic eddies, one of which (CE1) is the same we encountered
during the in-situ measurements. CE1 was the stronger as indicated with negative SSHA
between5°-9°N with core at 6.5°N latitude and is observed to be propagating from 86°E to 91°E
within one month (November to December).The eddy intensity is more in peak months i.e. in
December and January with a negative value of -0.12 m. CE1 propagates eastward to Andaman
waters and in December it is observed at 92°E. It begins to retract from Andaman waters by the
end of January and it is completely replaced by a positive sea surface. But the low is observed in
Bay of Bengal waters even during February centered at 87°E shifted northwards to 9°N.  The
shape of eddy is elliptical with its axis oriented in east west direction. The map also showed a
positive SSHA oriented in east west direction in the north of CE1. The eddy CE1 characteristics
and generating mechanism is described in the above section using in situ as well as satellite
observations.
The SSHA maps also revealed a cyclonic eddy located at 13°N and 88°E during November with
negative anomaly of -0.07 m. This eddy is marked as CE2.  Another eddy, CE3 is noticed at
13°N and 93°E which is comparatively of strong intensity than the CE2 (eddy at 88°E).  In
December, the shape of CE2 became elliptical with its axis oriented in an east−west direction
along 88°E. The negative anomaly is more in November with a SSHA of -0.12 m and the



intensity decreased during December with SSHA of -0.05 at two cores at 88°E and 93°E. CE3is
departed from western coast of Andaman to Bay of Bengal region during January with high
negative anomaly was replaced by a low value of -0.005. Negative anomaly is replaced by
positive anomaly very near to the western side of the Andaman Island. By the month of
February, it is completely departed from the Andaman waters. The CE1, CE2 and CE3 are meso-
scale features with diameter varying from 50-250Km.
Having recognized eddies from SSHA maps, further we have confirmed the prevailing processes
to the surface temperature and chlorophyll.  Cyclonic eddies due to the divergent forcing at the
center is occupied with sub-surface nutrient rich waters at the core and these area of negative
SSHA will be of relatively cool SST and high chlorophyll concentration as compared to other
regions.
SST is high in the initial phase of wintermonths i.e. in November (Fig.8a) with higher value
existed in entire region of Andaman waters (28.2°C-28.8°C).  During December (Fig.8b),
however, the values changed to 27.6°C-28.8°C. Further during January (Fig. 8c) and February
(Fig. 8d) the basin wide temperature was in the range to 27°C-29°C and 26°C-29°C respectively.
Though the Andaman waters were warm in general, the cold core eddies identified show
relatively cool temperature due to the prevalent cyclonic flow associated with it.  CE1 records
temperature 28.6°C during Nov, and when the eddy advances to the Andaman waters the surface
temperature begins to cool. SST decreases from 28.6°C to 28.2°C during December. SST again
decreased to 27.6°C in January. But in February the temperature remains the same as in the case
of January. CE2 also shows warm temperature during November (28.8°C) and decreases to
27.8°C in November. The decreasing trend follows in January also (27°C). The SST remains the
same in February also (27°C). CE3 displays the temperature of 28.6°C during November. During
December temperature decreases to 28.2°C and it again decreases to 27°C during January and
again decrease during February (26.5°C). The increased temperature in the eastern Andaman
might be due to the intrusion of low saline waters through Malacca strait (28-29°C, 32.3-34) as
inferred by Rama Raju et al., 1981 and Tan et al., 2006
High chlorophyll concentration is expected in eddy region due to enhancement of nutrients at
surface.  This cold core eddies are important because it is the area of high biological activity.
These areas are observed with strong physical and biogeochemical coupling resulting high



chlorophyll concentration. Generally, Andaman waters are oligotrophic in nature with less
chlorophyll concentrations (Vijayalakshmi et al., 2010). The existence of cyclonic circulation
increases chl a level in the eddy region.  When the cyclonic flow advances, the increased chl a
level was observed in the eddy locations at CE1, CE2 and CE3. CE1 records 0.1 mg m$^{-3}$during
November and it increased to 0.8 mg m$^{-3}$during December and decreased to 0.3 mg m$^{-3}$ January
(Fig. 7a-d). Chl a level decreased to 0.2 mg m$^{-3}$in February. CE2 displays lower chl a (0.2 mg m$^{-3}$)
in November. It increased to 0.8 mg m$^{-3}$during December and decreased to 0.6 mg/m$^3$ in January.
It shows a lower value of 0.2 mg m$^{-3}$ in February. CE3 revealed a very low value (0.1 mg m$^{-3}$)
during November.  During December, the chl a begin to increase in the eddy region (0.4 mg m$^{-3}$)
and in January also the pattern follows with a concentration of 0.4 mg m$^{-3}$and decreased to 0.2 mg
m$^{-3}$in February.
The role of wind stress curl on inducing the eddy is verified with weekly progress in the wind
stress curl (ASCAT) for the pockets. At CE1 the curl varies from -4.43x $10^{-7}$ to 1.28x $10^{-6}$ Pa m$^{-1}$
but the mode of the signal is -1.47x $10^{-7}$ Pa m$^{-1}$. At CE2 the curl ranges from -1.38x $10^{-6}$ to 1.12x
$10^{-6}$ Pa m$^{-1}$ , signal mode is -2.15x $10^{-8}$ Pa m$^{-1}$. The wind curl at CE3 shows values between -
2.87x $10^{-7}$ and 2.09x $10^{-6}$ Pa m$^{-1}$ and mode is -3.25x $10^{-8}$ Pa m$^{-1}$. However, the occurrence of
maximum negative values implies wind is not a dominant causative factor for the generation of
eddy.
As we described earlier, the role due to planetary wave for the eddy formation is analysed using
the Hovmoller plot of SSHA at 13°N and along 85°E to 93°E (CE2) [Fig.9]. The low SSHA
indicated the presence of upwelling mode Rossby wave in the region. It exhibits a continuous
westward propagation of a low SSHA signal along 13° N. This point out the existence of the
westward propagating Rossby waves in the region. The signal takes 80-90 days travelling from
the Andaman coast to the eddy core region at 88°E which have a phase velocity of 0.053 m s$^{-1}$ .
The theoretical phase speed of westward propagating signal at 13°N is calculated as 0.055 m s$^{-1}$.
The estimated speed is well compares with the theoretical speed (Jury and Huang, 2004). The
baroclinic instability due to westward propagating Rossby wave plays a dominant role in the
eddy generation and sustenance in Andaman and Bay of Bengal.
At CE3 the surface temperature is low compared to nearby location (27-27.2°C) and the MLD is
also deep (>70m). Wind is northeasterly with magnitude 4 to 7 m s$^{-1}$. The specific humidity 14
to18g/kg implies the dry continental air during the period. Net heat flux varies from $^-$98 to $^-$134 W



m$^{-2}$ during November – February. This causes heat loss due to evaporation (latent heat flux-220-
312 W m$^{-2}$) resulting cooling in the sea surface. Solar radiation varies from 114 to170 W m$^{-2}$in
the eddy region. This low solar insolation reduces the SST and resulting densification of water.
Thus, the surface water sinks and nutrient rich water entrains from deeper depths. This evince
that the atmospheric forcing causes surface cooling and the resulting convective mixing entrains
nutrients into the upper layer which activates the primary production (Prasanna Kumar and
Prasad, 1996, Madhupratap et al., 1996).
**Conclusion**
The column dynamics, forcing mechanisms, chemical and biological responses of cyclonic
eddies is explained for the Andaman waters based on a suit of in situ and satellite datasets. The
processes are small scale in nature with 100-250 km diameter and are found to be induced as a
result of baroclinic instability arised due to the westward propagating Rossby wave, semi-annual
mode with phase speed 0.20 m s$^{-1}$ and 0.55 m s$^{-1}$ respectively for CE1 and CE2, while CE3
associated with the process of convective mixing process occurring in the region due to cold dry
continental air from north east. The study put forward that, in addition to the mesoscale
processes triggering biological production, the convective mixing occurring along the North-
west coast of Andaman is taking a substantial role, though limited to a narrow strip along the
coast. The substantial increases in the regional surface biological production indicate the
complementary role of such processes in bringing up the quality of production in Andaman
waters. The role of convective mixing and eddies in the dynamics of the Andaman waters are
explained for the first time through this study.
**Acknowledgements**
Authors are grateful to the Ministry of Earth Sciences for supporting
the work and for providing facilities onboard FORV Sagar Sampada for in situ
measurements. All the fellow participants of the cruise FORV SS292 are
thankfully acknowledged. In situ data are obtained from FORV Data Centre in CMLRE.
ASCAT Scatterometer wind field is obtained from NOAA/NESDIS. The TOPEX/Poseidon
SSHA product is generated from the Merged Geophysical Data Record. Chlorophyll data was
retrieved from GSFC NASA. Heat flux data is provided by WHOI OAFlux project.

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

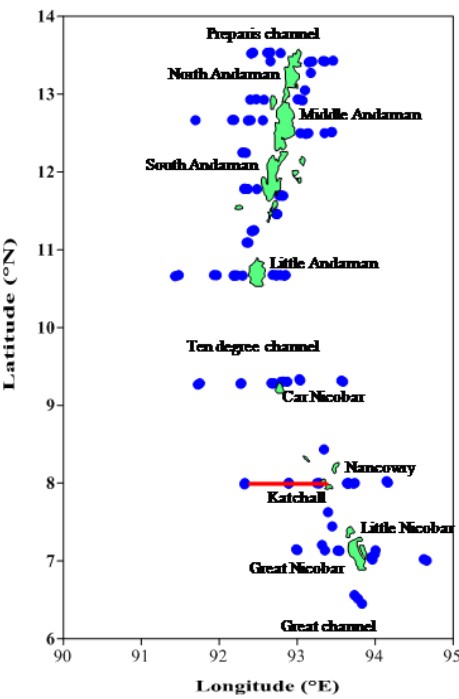

Fig. 1 Station Location

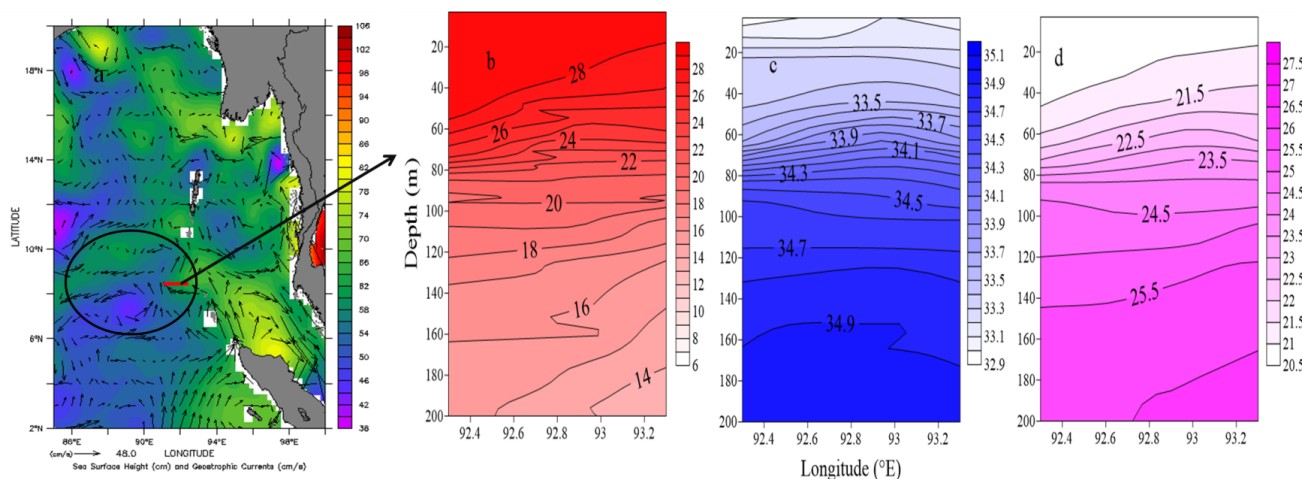





Fig. 2 a) Sea Surface Height and geostrophic current and the eddy location b) Vertical temperature, c) salinity and d) density distribution at the eddy location

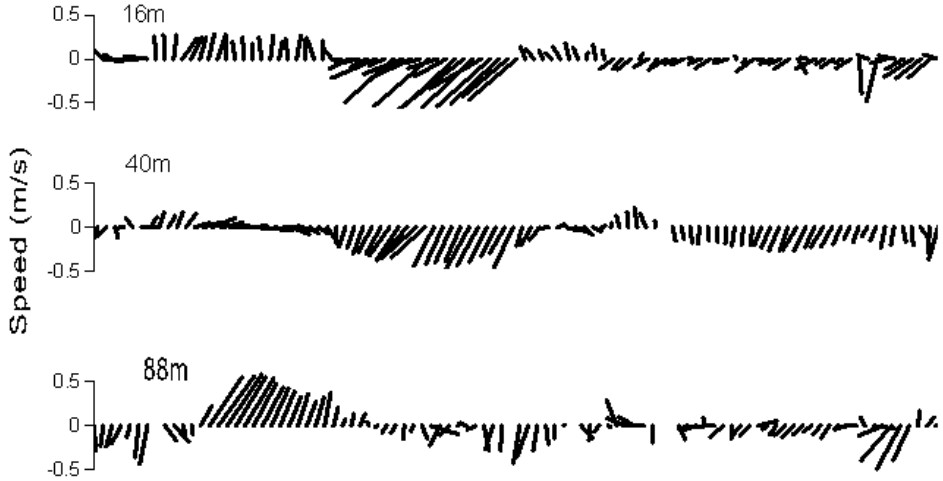

Fig. 3 Vertical current pattern along 8°N

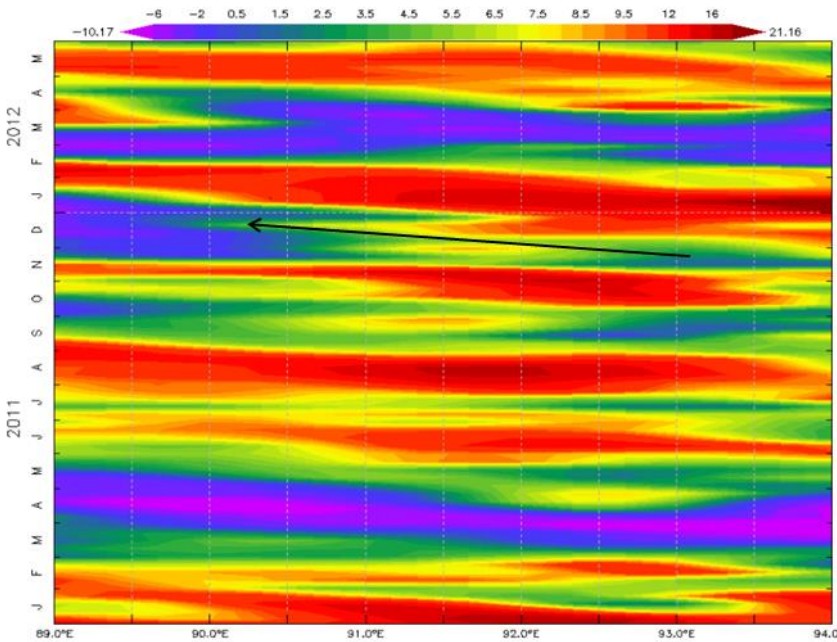

Fig. 4 Hovmuller diagram of SSHA along 8°N




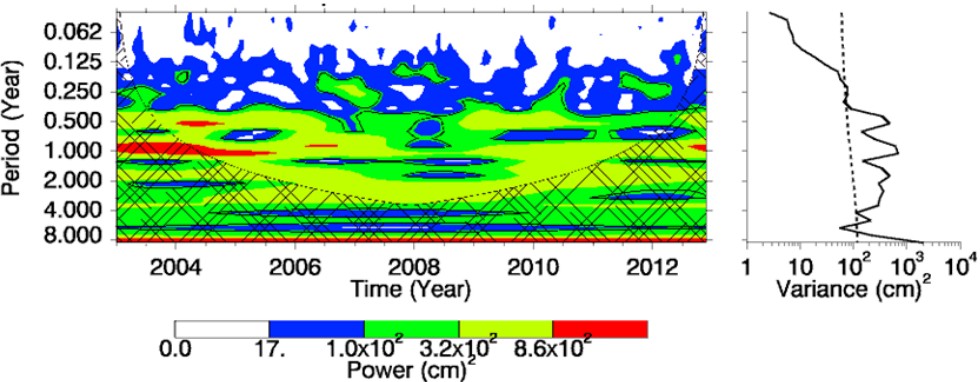

Fig. 5 wavelet power spectra of SSHA

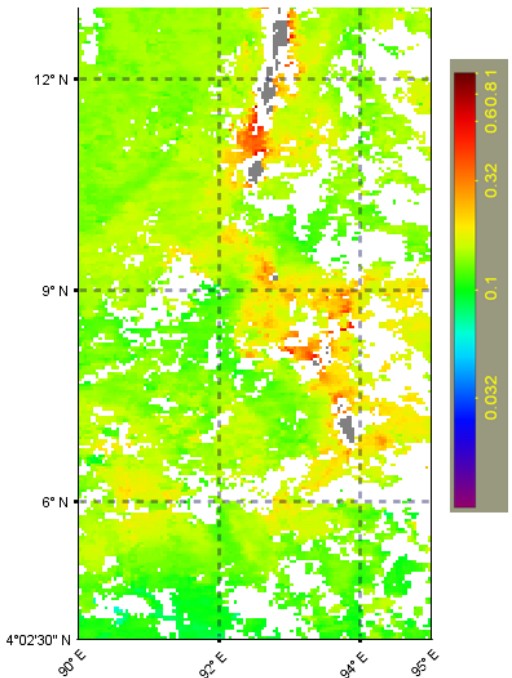

Fig. 6 chl a pattern during the in situ observation



Fig. 7 SSHA during a) November, b) December, c) January, d) February





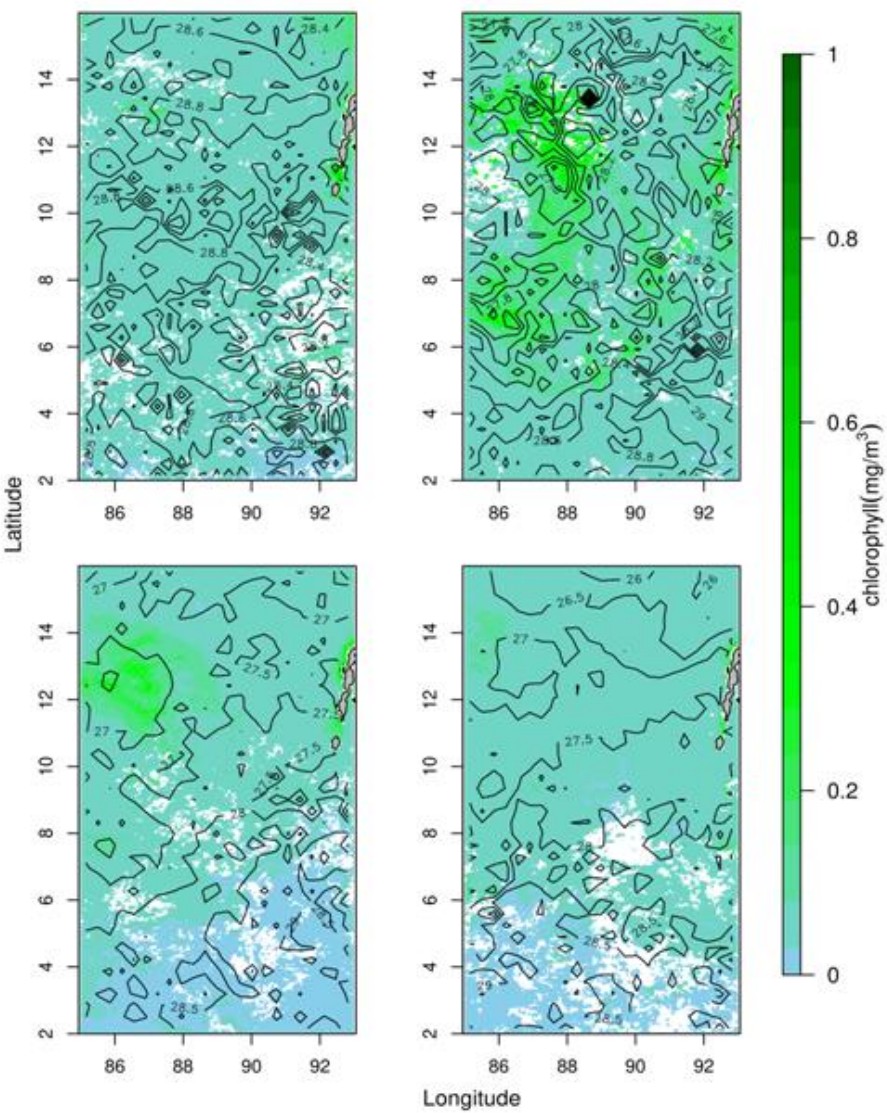

Fig.8 Overlap map of SST and Chl a during a) November, b) December, c) January, d)  February



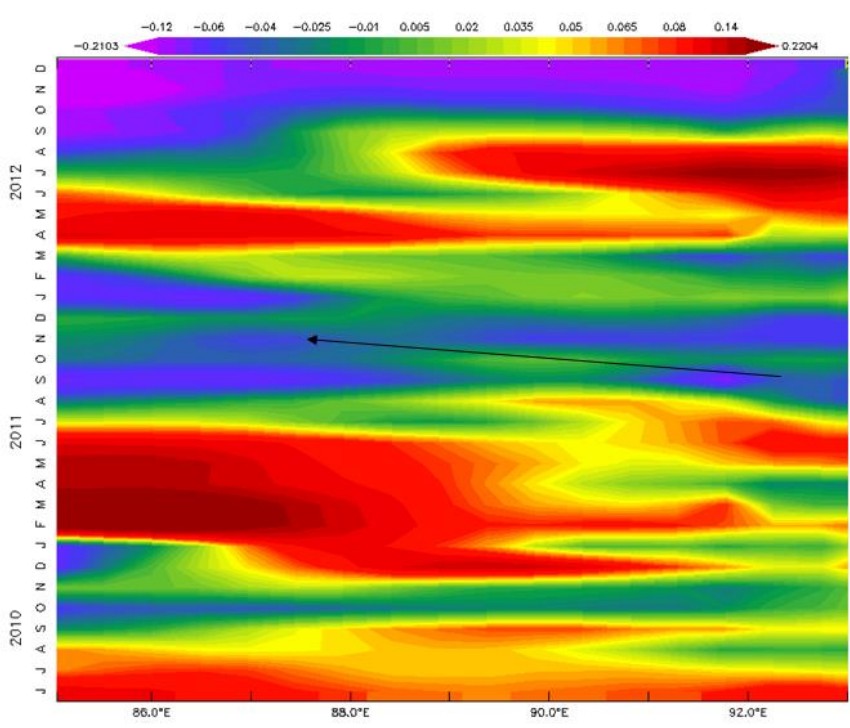

Fig. 9 Hovmuller of SSHA along 13°N

Table. 1 . Distribution of DO, NO₃, PO₄ & SiO₄ in the eddy region

| Lat.(°N) | Long.(°E) | Depth (M) | Depths | $NO_3$ µM | $SIO_4$ µM | $PO_4$ µM | DO (ml/L) |
|---|---|---|---|---|---|---|---|
| 8.00 | 92.33 | 1132 | 0 | 0.67 | 0.77 | 0.12 | 4.38 |
|  |  |  | 10 | 0.31 | 0.52 | 0.09 | 4.69 |
|  |  |  | 20 | 0.22 | 0.36 | 0.06 | 4.41 |
|  |  |  | 30 | 0.16 | 0.19 | 0.09 | 4.56 |
|  |  |  | 50 | 0.14 | 0.10 | 0.11 | 4.33 |
|  |  |  | 75 | 9.05 | 13.59 | 0.66 | 1.85 |
|  |  |  | 100 | 18.26 | 21.05 | 1.02 | 1.09 |
|  |  |  | 120 | 20.01 | 21.98 | 1.10 | 2.03 |
|  |  |  | 150 | 24.62 | 15.48 | 0.88 | 0.64 |
|  |  |  | 200 | 27.66 | 25.17 | 1.29 | 0.51 |
|  |  |  | 300 | 31.96 | 35.47 | 1.54 | 0.27 |



| | | | 500 | 37.46 | 44.01 | 1.58 | 0.23 |
|---|---|---|---|---|---|---|---|
| | | | 750 | 38.72 | 67.99 | 1.75 | 0.71 |
| | | | 1000 | 31.83 | 78.11 | 1.74 | 1.00 |
| 8.00 | 92.89 | 1052 | 0 | 0.59 | 0.18 | 0.11 | 4.79 |
| | | | 10 | 0.14 | 0.13 | 0.09 | 4.60 |
| | | | 20 | 0.62 | 0.69 | 0.09 | 4.93 |
| | | | 30 | 1.32 | 1.72 | 0.13 | 4.56 |
| | | | 50 | 4.14 | 6.43 | 0.30 | 2.96 |
| | | | 75 | 10.02 | 16.54 | 0.68 | 2.02 |
| | | | 100 | 14.00 | 18.17 | 0.84 | 1.57 |
| | | | 120 | 19.12 | 24.94 | 1.12 | 0.91 |
| | | | 150 | 22.02 | 25.82 | 1.20 | 0.81 |
| | | | 200 | 25.07 | 31.29 | 1.36 | 0.60 |
| | | | 300 | 29.18 | 35.62 | 1.50 | 0.36 |
| | | | 500 | 32.10 | 45.72 | 1.60 | 0.40 |
| | | | 750 | 34.20 | 64.89 | 1.79 | 0.75 |
| | | | 1000 | 37.22 | 81.82 | 1.80 | 1.25 |
| 8.00 | 93.25 | 215 | 0 | 0.98 | 0.44 | 0.15 | 4.73 |
| | | | 10 | 0.21 | 1.90 | 0.14 | 5.00 |
| | | | 20 | 0.24 | 1.94 | 0.16 | 4.71 |
| | | | 30 | 2.03 | 6.69 | 0.30 | 3.85 |
| | | | 50 | 8.01 | 13.89 | 0.63 | 2.49 |
| | | | 75 | 10.02 | 18.88 | 0.88 | 1.70 |
| | | | 100 | 17.04 | 25.36 | 1.07 | 1.42 |
| | | | 120 | 24.09 | 27.26 | 1.20 | 1.37 |
| | | | 150 | 28.35 | 31.88 | 1.38 | 0.79 |
| | | | 200 | 31.01 | 36.47 | 1.47 | 0.63 |
| 8.00 | 93.28 | 100 | 0 | 0.83 | 0.64 | 0.18 | 4.63 |
| | | | 10 | 0.07 | 0.48 | 0.15 | 5.37 |
| | | | 20 | 1.08 | 1.20 | 0.18 | 4.49 |
| | | | 30 | 1.78 | 2.89 | 0.23 | 4.46 |
| | | | 50 | 4.65 | 5.69 | 0.30 | 4.00 |
| | | | 75 | 15.38 | 17.09 | 0.66 | 2.25 |
| | | | 100 | 23.50 | 20.87 | 0.96 | 1.49 |
| 8.00 | 93.29 | 68 | 0 | 0.71 | 1.62 | 0.27 | 4.71 |
| | | | 10 | 2.14 | 6.50 | 0.39 | 4.85 |
| | | | 20 | 2.31 | 7.14 | 0.35 | 4.57 |
| | | | 30 | 3.19 | 7.86 | 0.39 | 4.48 |
| | | | 50 | 5.05 | 8.64 | 0.45 | 3.59 |