# Peer review of "Mesoscale processes regulating the upper layer dynamics of Andaman waters during winter monsoon"

_Ocean Science, 2018_

## Referee Comment (RC1) · N. Vissa (Referee) · 6 Jul 2018

Authors aims to understand the mesoscale processes of the upper ocean dynamics of Andaman waters during winter monsoon. In the present study authors have used the satellite altimeter, winds and in situ FORV Sagar Sampada crusie data sets.

Technical suggestions as follows.

1) Authors have mentioned the mixed layer depth and isothermal layer depth. It would be better to mention the method or criteria adopted in the manuscript.

2) Authors need to discuss about the role of ENSO in modulating the oceanic eddies

and planetary waves in the discussion section.

3) There are several methods to quantify the oceanic mesoscale eddies (e.g Okubo-Weiss) , please adopt any objective method, such that eddy identification and tracking can be done.

4) Wind stress curl values over the locations can be provided in a tabular format.
* * *

---

## Referee Comment (RC2) · Anonymous Referee #2 · 3 Sep 2018

[]article geometry a4paper

[Figure]

**Comments on the paper entitled "Mesoscale processes regulating the upper layer dynamics of Andaman waters during winter monsoon.**

September 3, 2018

The manuscript aims to characterise a set of mesoscale eddies in the Andaman Sea and relate their formation to the incidence of Rossby Waves propagating within this basin. The authors have done an extensive amount of work, analysed a variety of datasets, and reached several conclusions. It is clear that the authors know several analysis tools and have technical skills. However, there is no cohesion in the manuscript.

I noticed that there are several separate analysis, tied into one story in the manuscript. However, this story still requires a lot of work. Most of the time, the authors are describing events of different spatial scales and trying to relate them (e.g., sub-mesoscale coastal processes and mesoscale oceanic eddies). I could not identify the knowledge gap this paper is trying to fill. I suggest the authors to identify the key message they want to convey to the reader - and to clearly identify in the manuscript what are the main contributions to their field. Not all the analysis done in a project have to be included in a manuscript. The results contained in the manuscript should all act as a

support for achieving the goal proposed in the Introduction.

I suggest major revisions are made before this manuscript can be considered again for submission. I truly hope that my comments help.

**1 Major Comments**

1. Observation-based analysis (Figs 2 and 3): I'm not convinced you sampled an eddy in the stations you show. Yes, there is some northward flow to the west and some southward flow to the east, but it is hard to confirm it is an eddy on not only a current interacting with the continental shelf. The SSH map cannot confirm this is an eddy - it does not resolve this scale (only features that are larger than 110 km in radius)

2. Even if you still think that the ADCP data and the SSH maps indicate the presence of a sub-mesoscale eddy, this eddy is anticyclonic (clockwise rotation in the northern hemisphere). Therefore, all the discussion and data interpretation that points this eddy as being cyclonic is erroneous (e.g., lines 144, 152). Please, re-interpret your data keeping this in mind.

3. I could not understand why the authors show the vertical sections of T, S, and density only down to 200 m in Figure 2. If these variables where sampled by the CTD, I would expect deeper measurements. If you look at values below 200 m, you might get more insight about the structure you sampled.

4. Still about Figure 2: the max and min values in the colour axes in (b), (c), and (d) are not appropriate. This choice might be hindering some isolines above 40 m. Please review this figure.

5. I could not understand the advances this manuscript brings to the Rossby Wave propagation and eddy triggering to the literature. Please state it in the manuscript.

6. I was not convinced that the Rossby Waves indeed triggered the eddy. You need more results to support this claim. The whole section on "Generation Eddy Mechanism" needs through review and more results to confirm your claims. The statement in lines 218-220 needs proof to be accepted.

7. Figure 6 shows higher [chla] close to the islands. You claim this is because of eddy effect. It just looks like it is a natural coastal increase in nutrients (river runoff, upwelling, current-bottom friction). Is this region of the world different, and these processes would not be in place?

8. Figure 8 actually suggests an increase in [chla] caused by the presence of an eddy. See the spiralling green patch between 11-14N and 85-88E. This might relate to a cyclonic eddy.

9. The part of the manuscript that requires a specially careful analysis and interpretation is the SSHA analysis (Figure 7). The SSH product used does not resolve the features you are trying to investigate. You need to zoom out to look at the mesoscale eddies. In addition, an eddy is defined by a closed SSH contour. You cannot see this in any of the features you indicate as "eddies". All the paragraphs in the manuscript related to this figure must be intensively corrected.

10. I could not comment on the biogeochemical results (Table 1) and in the wind stress results because they are not presented in a suitable manner. Please make a figure with the values in Table 1 and a figure with the wind results if included in the next manuscript.

11. The domains you look at in Figures 1, 2, 6, 7, and 8 are all different in space - and probably in time (but I can't tell because this information is not given). You cannot

discuss the "eddies" from these different datasets as you do here because they are not the same ones!

**2  Minor Comments**

1. Latest manuscripts in our field are written in active voice - it is more engaging for the reader and it helps to transmit your information clearly.

2. Choose a verbal tense and stick to it. Usually, the manuscript is written in the present tense, with Data and Methods section written in the past tense.

3. The goal of the manuscript is missing in the abstract

4. The usage of the term "vertical current" is wrong (see lines 8, 139, 175 for example). This term relates to the w component of the current - which is not the case here. Please re-phrase.

5. The introduction needs to be re-structured. Usually, Introductions go from the "big scenario", to the "small scenario", to the knowledge gap you are trying to fill.

6. Lines 42-44 ("Explanations have come ..."): Is this the contribution of this manuscript or something reported in the literature? This is a case where the use of active voice helps.

7. Lines 50-52: does this relate to Andaman eddies or eddies in general?

8. Lines 61-63: does this relate to Burnaprathepart 2010 eddy or to the eddy you describe in this manuscript?

9. In the Data and Methods section, you should only include what you used in your manuscript. For example, you did not analyse AVISO data between 2003 to 2013.

[Figure]

You only showed some certain days. You also did not use the meteorological information (lines 72-74) and solar radiation information (lines 97-99) here, so no need to say you collected it.

10. Lines 86-88: this paragraph does not inform the reader about the local bathymetry. you should remove it and add the bathymetry in Figure 1.

11. Please describe in the manuscript the reason for working with the weekly AVISO dataset, instead of the daily product.

12. Line 107: Before describing the wavelet analysis, it helps the reader if you write a brief line saying what you use it for later on the manuscript.

13. Line 122: Physical characteristics of an eddy is not the same as eddy dynamics. You do not approach eddy dynamics in this manuscript.

14. Line 130: missing citation to paper that defined Bay of Bengal waters

15. Line 151: The figure does not show this is a sub-surface cyclonic eddy. You are not resolving this feature neither in the horizontal direction or in the vertical direction.

16. Line 244-247: The fact that the eddies are propagating eastwards are not shown in your figure - and is also hard to agree. Eddies tend to propagate westwards, unless when advected by a strong flow.

17. Line 254: anticyclonic eddy

18. Lines 307-316: This paragraph again describes results related to Rossby Waves. Could you merge this paragraph to the previous one that discusses Rossby Waves? This would help readability. In addition, merge Figures 4 and 9 and discuss them together.
**3  Figures and Table**

- Figure 1: Please add a larger map indicating the location of this archipelago in context; add the regional bathymetry.

- Figure 2: The SSH data shown in (a) does not resolve your feature shown in (b-d). You can still show this SSH map, just to show the oceanographic context of the region at the time of sampling - but zoom out to show more - and a map of SSHA better shows mesoscale eddies; your circle in (a) does not coincide with your sampling location; say when is this SSH map from; please review isolines and axis limits in (b-d); if data is available below 200 m, show it - if not, say in the manuscript why you stop at 200 m; show the location of the stations in (b-d) for the reader to know how much interpolation is happening in this figures.

- Figure 3: This is not vertical current; these are horizontal (or maybe only meridional or zonal?) currents at three different depths; add x-axis indicating longitude.

- Figure 4: indicate which data you use to build this diagram (Aviso); merge this figure to Figure 9; include units.

- Figure 5: indicate which data you use to build this diagram;

- Figure 6: Is this data from one day? Is this a composite? indicate where is this data from, and the time period shown.

- Figure 7: Are these from one day for each month? Are these means? Indicate where data is from; you cannot say that your CE3 and CE1 labels relate to eddies, because you are not showing the closed contours - zoom out to check; label CE2 might relate to a very small anticyclonic eddy (positive SSHA and clockwise rotation in the NH) - not cyclonic.
- Figure 8: Are the SST values shown here for one day? are these combined maps of chla? The T values are very hard to see - increase label font.

- Table 1: This table does not help the reader and does not provide any useful information as it is - please make a figure to show these values; specify "eddy region" in the caption.

---

## Referee Comment (RC3) · Anonymous Referee #3 · 10 Sep 2018

Manuscript entitle "Mesoscale processes regulating the upper layer dynamics of Andaman waters during winter monsoon" by Chandran et al., used multiple data set to study eddy in the Andaman Sea. It is an interesting study but identification of eddy is through qualitative approach. I have few suggestions on their manuscript

(1) Oceanic circulation in upper ocean is not only geostrophic, identification of eddy through geostrophic current is not enough. It is better to follow Okubo-Weiss parameter method to identify eddies. (2) It is suggested to use OSCAR current observations rather than using geostrophic current. (3) It is known that equatorial westerly jet (Wyrtki jet) produces down welling Kelvin wave which propagate as a coastal Kelvin and radiate Robby wave which favors cyclonic eddies in the BoB including Andaman Sea. These mechanism is already reported by earlier study but here authors connected it with bio-geochemistry using in-situ should be reflected and discussed in detail. Author mentioned that they measured vertical velocity but throughout the manuscript it is not displayed. It is suggested use them in this study and may compare it with up welling of isotherms etc. (4) Author mentioned vertical stability and vertical shear of currents favors the formation of eddies. It can control vertical mixing, however shear of horizontal current in horizontal plane supports the formation of eddies. This what discussed by Okubo-Weiss. Author should drop related sentences from the abstract about vertical shear and stratification. (5) It is mentioned that BoB upper ocean stratification restrict nutrient supply and later it is mentioned that eddy support up welling and nutrient supply, however it is not clear what is a role of waves and convective mixing? According to figure 8 higher chlorophyll is reported in north BoB, where stratification is more.

Minor comments: (1) Figures quality need to be improve and better display eddy location in spatial plots of all parameters. (2) Figure captions are short i.e "wavelet power spectra of SSHA", and contain little information about figure. (3) Figure 3 X axis caption is missing. (4) Figure 2 caption should be "depth longitude section of b) temperature, c) salinity and d) density distribution at the eddy location" than "b) Vertical temperature, c) salinity and d) density distribution at the eddy location". (5) In Figure 2 data used in the present study along the ship track is displayed, whereas in figure 1 other data locations are also mentined which are not used in the manuscript. Better to drop figure 2 and all the points from where data collected and used in the present study can be displayed in Figure 2a. (6) Figure 6 include eddy location in this plot. (7) Figure 4 and 9 can be merged as 4a and 4b (8) In Figure 7 same period OSCAR current should be overlay. (9) Figure 8 (a-d) are missing in figure, confusing to know it.

---

## Author Comment (AC1) · 13 Dec 2018

Reviewer 1 1) Authors have mentioned the mixed layer depth and isothermal layer depth. It would be better to mention the method or criteria adopted in the manuscript Included the same in the revised manuscript 2) Authors need to discuss about the role of ENSO in modulating the oceanic eddies and planetary waves in the discussion section.

Chen et al. (2012) studied the interannual variability mechanism of the mesoscale eddies in BoB and pointed that the eddy activities do not directly link to El Nino Southern Oscillation (ENSO) events and are sensitive to the baroclinic instability of the back-

ground flow.

3) There are several methods to quantify the oceanic eddies (e.g Okubo-Weiss), please adopt any objective method, such that eddy identification and tracking can be done.

Adopted Okubo-Weiss parameter method to track the eddy and included in the revised manuscript.

4) Wind stress curl values over the locations can be provided in a tabular format

By adopting the Okubo-Weiss method we identified only one eddy. The wind stess curl values for CE2 and CE3 dropped from the revised manuscript

Please also note the supplement to this comment:
https://www.ocean-sci-discuss.net/os-2018-23/os-2018-23-AC1-supplement.pdf

[Figure]

[Figure]

Fig. 1 Station Location

**Fig. 1.** Fig. 1 Station Location

[Figure]

Fig. 2 a) Sea Surface Height (cm- Aviso weekly) and geostrophic current (cm/s) and the eddy location   b) Vertical temperature (°C), c) salinity and d) density (kg/m³) distribution at the eddy location

**Fig. 2.** Fig. 2 a) Sea Surface Height (cm- Aviso weekly) and geostrophic current (cm/s) and the eddy location b) Vertical temperature (ïĆřC), c) salinity and d) density (kg/m3) distribution at the

[Figure]

Fig. 3   Horizontal current (m/s) structure at different depth at 8°N

**Fig. 3.** Fig. 3 Horizontal current (m/s) structure at different depths along 8°N

[Figure]

Fig. 4  Hovmuller diagram of SSHA(m) (Aviso monthly) along 8°N

**Fig. 4.** Fig. 4 Hovmuller diagram of SSHA(m) (Aviso monthly) along 8°N

[Figure]

Fig. 5. Wavelet spectra of SSHA (m- Aviso monthly from 2003-2013) along 8°N

**Fig. 5.** Fig. 5. Wavelet spectra of SSHA (m- Aviso monthly from 2003-2013) along 8°N

[Figure]

Fig. 6 chl a (mg/m³- weekly MODIS Aqua) pattern during the insitu
observation

**Fig. 6.** Fig. 6 chl a (mg/m3- weekly MODIS Aqua) pattern during the insitu observation

[Figure]

Fig. 7 Merged map of SSHA (m), Geostrophic current (cm/s) and Okubo-Weiss paremeter (Black contour of -2x10-11/s2) from Aviso during a) November b) December c) January d) February

**Fig. 7.** Fig. 7 Merged map of SSHA (m), Geostrophic current (cm/s) and Okubo-Weiss pareme-ter (Black contour of -2x10-11/s2) from Aviso during a) November b) December c) January d) February

[Figure]

Fig.8 Overlap map of SST (°C-monthly MODIS Aqua) and Chl a (mg/m³- monthly
MODIS Aqua) during a) November, b) December, c) January, d) February

**Fig. 8.** Fig.8 Overlap map of SST (ïČřC-monthly MODIS Aqua) and Chl a (mg/m3- monthly
MODIS Aqua) during a) November, b) December, c) January, d) February

[Figure]

**Supplement:**

**Mesoscale processes regulating the upper layer dynamics of Andaman waters during winter monsoon**

*Salini.T.C[1], Smitha.B.R[2], Sajeev.R[1], Lix John. K[1], Midhunshah Hussain[1], Rafeeq.M[2]

Cochin University of Science and Technology, Kochi, 682016, India

Centre for Marine Living Resources & Ecology, Kochi, 682037, India

*Corresponding Author. email: salinitc@gmail.com, ph.+91 984688249

**1 Abstract**

The characteristics of cold core eddies and its influence on the hydrodynamics and biological
production in Andaman waters were studied using insitu and satellite observations. The specific
structure and patterns of the temperature-salinity (T-S) profiles, nutrients and chl a indicate the
occurrence of the eddy, the spatial extent of which is well marked in sea surface height anomaly
(SSHA). The Cyclonic Eddies are tracked using Okubo-Weiss parameter of $^{-}2x10^{-11}$ /s$^2$ centered
at 8°N and 92°E, and 13°N and 93°E (CE1 and CE2 respectively). Insitu measurements are done
in the eastern flank CE1 along 8°N and 92.5-93.5°E. Vertical currents recorded using Acoustic
Doppler Current Profiles (ADCP) shows northward flow along the track (0.3m/s) while along the
western flank, the flow is weak and southward. This evidence the occurrence of cyclonic eddy
and the altimetry derived SSHA depicts the spatial extent. Analysis to explore the possible
forcings to induce the occurrence of eddy, indicate baroclinic instability (Ri <0.0001) in the
water column due to vertical shear in the horizontal flow. The presence of Bay of Bengal (BoB)
water in the region as evidenced in the T-S profiles, and the presence of semiannual Rossby
waves in the region accounts the contribution, whereas, wind stress curl was not a major
inductive of divergence in the region. Though less significant, the eddy is formed to influence
the nutrient pattern (NO$_2$, NO$_3$, PO$_4$ and SiO$_4$) and the biological production (chl a). The eddy
influenced the nutrient pattern (NO$_2$, NO$_3$, PO$_4$ and SiO$_4$) and the biological production (chl a) in
the region. CE2 is associated with convective mixing processes occurring along the northwest
coast of Andaman due to the prevalent cold dry continental air from north east.

**21 Introduction**

The Sea around the Andaman and Nicobar Island chain is influenced by reversing
monsoon with moisture rich summer winds and dry continental air flow from north-east during
winter (Potemra et al., 1991). The region receives enormous runoff and suspended matter from
Ayeyarwady-Salween river system, which has significant influence on the hydro-dynamics and
oceanography (Robinson et al., 2007). The region is characterised by strong stratification,
prevents vertical mixing, causes nutrient depletion in the upper layers and subsequently leads to
oligotrophy. The seasonal winds, moderate or strong, though are experienced during the
summer and winter months, are not found to exert any divergence or positive curl and nutrient
pumping to enrich biological production is least encountered in these waters. The sea is less
productive compared to the Arabian Sea and Bay of Bengal and average primary production
during fall inter-monsoon is 283.19 mg $C/m^2/d$ followed by spring inter-monsoon (249 mg
$C/m^2/d$), summer monsoon (238.98 mg $C/m^2/d$) and winter monsoon (195.47 mg $C/m^2/d$)
[Sanjeevan et al., 2011].  Earlier observations show that the eastern and western part of the
island chain is governed by distinct water properties where west shows typical BoB
characteristics, northeast is highly influenced by the Ayeyarwady and Salween river system and
the southeast by the productive environment of Malacca strait (Salini et al., 2010). The region is
least explored for oceanic processes and surveys conducted so far for understanding the
biodiversity and the basin scale environment associated with the living resources indicate the
absence of any major or seasonal processes that result in nutrient pumping to alter production
pattern. However, the emergence of satellite techniques, especially the Altimetry and Ocean
Color imageries on mesoscale to basin scale, the understanding of the upper layer dynamics has
been strengthened. Explanations have come on such major processes in the BoB, especially on
number of eddies and gyres and also the impact of cyclones, which causes enormous mixing in
its path (Nuncio and Prasanna Kumar, 2012). Eddies are mesoscale processes (50–200 km
diameter) and ubiquitous feature of the ocean occurs in both clock-wise and anti-clock wise
direction, resulting in convergence/divergence at the center.

Mesoscale eddies play a dominant role in transportation of salt, heat and nutrients within the
ocean (Dong et al., 2014) and enhance local production in oligotrophic areas (Hyrenbach et al.,
2006), ultimately influencing the production pattern in each trophic level (Bakun, 2006).
Mechanisms behind the eddy formation has been suggested by many researchers; different
driving mechanisms have been attributed to eddy formation, such as Ekman pumping and remote forcing from the equatorial Kelvin wave reflecting off the eastern boundary as Rossby wave. According to Yu et al. (1999), westward propagating Rossby wave, excited by the remotely forced Kelvin wave, contribute substantially to the variability of the local circulation in ocean. Using the multilayer model, Potemra et al. (1991) described coastal Kelvin wave, which originates at the equator and propagates around the entire western perimeter of the region around both the Andaman Sea and the Bay of Bengal. Mesoscale eddies are observed in the coastal waters of the Andaman and Nicobar Islands (Hacker et al. 1998 and Chen et al. 2013) based on in situ hydrographic measurements. Burnaprathepart et al. (2010) described the presence of eddies in Andaman Sea and its role in enhancing the primary productivity synthesizing number of vertical profiles on chl a, major nutrients, temperature, as well as salinity. However, there are no comprehensive study undertaken for this region to explain the role of eddies (cold and warm cores) in the Andaman waters as a whole in regulating the available biological production. In this context it is attempted to enumerate these mesoscale processes based on SSHA imagery and geostrophic current pattern along with in situ evidences. The objective of this present study is to identify such processes in the basin, to explain the forcing mechanism and its response in column dynamics as well as biogeochemistry.

**Data and Methodology**

In situ measurements were taken onboard FORV *Sagar Sampada* during 21 November – 14 December 2011. The environmental characteristics are understood from station based measurements in the east and west of the island chain. However, the focus was to obtain a transect with 4 stations (Fig. 1) along the eddy. The meteorological parameters like air temperature, air pressure and humidity were also collected through the instruments/sensors attached to the IRAWS onboard in 15 minute interval. Profiles of temperature, salinity, dissolved oxygen and Sigma-t were obtained using SeaBird 911 Plus CTD with Niskin water samplers and deck unit for data acquisition. The datasets are processed for 1m bins. Salinity is also derived from water samples collected through Niskin samplers and using Guildline 8400A Autosal Salinometer to validate the CTD derived data.. Twelve numbers of 10 liter Niskin water samplers were used to collect water samples from standard depths (surface, 10, 20, 30, 50, 75, 100, 120, 150, 200, 300, 500, 750 and 1000 m) for measurements of dissolved oxygen and nutrients. Temperature-Salinity profiles for water mass characteristics are based on averaged (climatological) data from Levitus et al. (1994). Mixed Layer Depth (MLD) is derived from CTD profiles as the depth at which the seawater density (Sigma-t) exceeds the surface density by 0.2 kg/m$^3$ (Sprintall and Tomczak, 1993). The Isothermal Layer Depth (ILD), the depth of the top of the thermocline, is defined as the depth at which surface temperature decreases by 1 °C from sea surface temperature (Kara et al., 2000 and Rao and Sivakumar, 2003). The thickness of the barrier layer is computed as the difference between ILD and MLD (Lukas and Lindstrom, 1991).

Monthly composite of the chlorophyll data is obtained from the Distributed Active Archive Center (DAAC) of National Aeronautics and Space Administration, NASA. Dissolved oxygen was measured by Winkler titration. Analyses of nitrite, nitrate, phosphate and silicate were performed using a Skalar Analyser.

Wind stress curl (daily) data used was taken from ASCAT processed by NOAA/NESDIS utilizing measurements from the Scatterometer instrument aboard the EUMETSAT Metop satellites with a spatial resolution of 25 km; chl a data was taken from MODIS Aqua Level 3 at a spatial resolution of 4 km, downloaded from Ocean Color Website and processed using SeaDas. SST was obtained from MODIS Aqua Level 3 at a spatial resolution of 4 km downloaded from Ocean Color Website, while SSHA data obtained with 7 day temporal resolution from AVISO for the period from January 2003-January 2013. The cold core eddy was recognized through SSHA with geostrophic current imagery obtained from https://oceanwatch.pifsc.noaa.gov, and was observed to be centered at 7° N and 90° E with current moving in cyclonic direction. Net heat flux, solar radiation, latent heat flux, and specific humidity were obtained from http://oaflux.whoi.edu.

The eddies are spotted using two ways, first method is using SSHA contours and geostrophic currents, calculated from the following geostrophic equations,

$$u = -\frac{g}{f}\frac{\partial h}{\partial y} \qquad (1)$$

$$v = \frac{g}{f}\frac{\partial h}{\partial x} \qquad (2)$$

Where u and v are the zonal and meridional components of geostrophic currents, g is the gravitational acceleration, f is the Coriolis parameter, x and y are longitudinal, latitudinal co- ordinates and h is the SSHA.

Second method is using the Okubo-Weiss parameter, OW (Okubo, 1970 and Weiss,

1991) and is defined as

$OW = s_n^2 + s_s^2 - w^2$ (3)

Where $s_n$ is the normal strain component, $s_s$ is the shear strain component and w is the relative vorticity.

$s_n = \frac{\partial u}{\partial x} - \frac{\partial v}{\partial y}$ (4)

$s_s = \frac{\partial v}{\partial x} + \frac{\partial u}{\partial y}$ (5)

$w = \frac{\partial v}{\partial x} - \frac{\partial u}{\partial y}$ (6)

If the vortex core is dominated by vorticity, the negative Okubo-Weiss are predictable in the vortex core.

Wavelet transform is an appropriate analysis tool to study multi-scale, non-stationary
processes occurring over finite spatial and temporal domain. In this study, the wavelet was
used to analyse time series data of oceanographic parameters that contain non-stationary
power at many different frequencies. This technique is used to decompose time series into its
frequency components based on the convolution of the original time series with a set of
wavelet functions, and if possible, determine both the dominant modes of variability, and
how those modes vary with time. It expands functions in terms of wavelets, which are
generated in the form of translations and dilations of a fixed function called the Mother
Wavelet. In the present study the wavelet is applied to explain the temporal variation of
SSHA in the eddy region to explore the life span and frequency of the processes during the
10 years.Meyers et al. (1993) used wavelet analysis to study the propagation of mixed
Rossby-gravity waves in an idealized numerical model of the Indian Ocean.

The phase speed for long baroclinic Rossby wave is given by $C = \frac{-gH_0\beta}{f^2}$, (7)

where g is the reduced gravity term (taken as 0.04 m s$^{-2}$ for the first baroclinic mode), $H_0$

is the thermocline depth (taken as an annual mean depth of 20°C isotherm derived from Levitus and Boyer, 1994), f the Coriolis parameter and $\beta = \frac{\partial f}{\partial \phi}$, where $\phi$ is the latitude.

**Results and Discussion**

**Physical characteristics of the Eddy region**

The region is characterized by warm (27.6–28 °C), humid (72–77%) air and wind is from northeast, suggesting the prevalence of northeast monsoon condition of magnitude in the range of 10–12 m/s with comparatively lower speed (10 m/s) in the western part and higher speed (12 m/s) in the eastern part of the eddy (referred to hereafter as CE1).

The SST varies in the range of 28.4-28.8 °C with lower temperatures near the coastal water compared to offshore; the surface salinity (33.00 psu) and density (20.40 kg/m$^3$) values, on the other hand, are similar in coastal and offshore waters. Regional water mass characteristics from temperature, salinity, and density profiles show that the area is occupied by

BoB waters with temperature ranging from 28.0-28.5 °C, salinity 33.2-33.8 psu, and density

20.6-20.8 kg/m$^3$ (Salini et al., 2018). Vertical temperature distribution along 8°N (Fig. 2b)

shows warm (>28.5 °C) and thick isothermal layer (~54 m) in the western part and a gradual decrease towards east (20 m). The most important feature in the thermal structure is the upsloping of isothermal layer, which is prominent in the subsurface (54–220 m) also, and the mixed layer depth (MLD) shoaled from west to east (47–19 m). Vertical salinity and density distribution show the presence of low saline (32.9–33.1psu) water in the upper 30 m, with an upsloping tendency (Fig. 2 c, d) as in the case of temperature. Similar pattern is reflected in density characteristics too.

The horizontal current structure at 8° N along 92.5° E to 93.5° E shows irregular current pattern from surface to 90 m (Fig. 3). Along the eastern part of the 100 km transect, major flow is towards south ($\cong$30 km), west to it with a narrow and weak northward flow, followed by major southward drift up to 40 m. However, the response to this irregular pattern is insignificant in the T-S profiles and so the eastern part of the transect (~60 km) is not considered for addressing the eddy. In the western flank, the northward and the subsequent flow towards south indicate cyclonic flow direction. The current recorded at 16 m depth is considered for near surface pattern and this shows the presence of a northern component with a magnitude of 0.3 m/s in the eastern part negligible speed in the western part, directed westward. But at 40 m the current magnitude decreases in the eastern flank (0.1 m/s) and increases in magnitude in the western flank (0.1 m/s) with direction changing from northeast to southwest. The current at 88 m also follows the same pattern, but magnitude changes from 0.5 m/s in the western part and 0.4 m/s in the eastern part. The upsloping in the T-S profiles concurrent to this confirms the feature as a subsurface cyclonic eddy. The flow in the eastern flank is towards north (0.3 m/s) and at west it is to the south (0.5 m/s). The data was analyzed for all 8 m cells up to 88 m depth and found to follow the same pattern as that of near surface but with a decreasing magnitude. The dataset was seen to contain spurious values below 88 m and hence discarded.

**Eddy Generation Mechanism**

The possible physical mechanisms that govern the eddy includes the wind stress curl, topographic instability, shear flows, baroclinic instability and the radiation of Rossby waves from poleward propagating coastal Kelvin waves etc. (White, 1977 and Kessler, 1990). Daily wind stress curl is examined to identify the local forcing that contributes to the formation and sustenance of the eddy. Curl of the eddy region from ASCAT wind data shows negative values in the range of $-5.6 \times 10^{-8}$ and $-8.24 \times 10^{-8}$ Pa/m, indicating convergence and hence the contribution due to wind stress curl is ruled out.

Other possible eddy generation mechanisms are differential mixing of region with the adjacent sea mainly through inflow from Malacca Strait and freshwater influx from adjoining rivers leading to strong density variations in the water column. This variation may reduce or enhance the mechanical effects in the form of eddy or meanders in the region. This is measured based on the estimated Richardson Number (Ri). According to Miles (1961), the flow is stable if Ri>0.25.

Ri is calculated as $Ri = \dfrac{N^2}{(\frac{\partial u}{\partial z})^2}$        (7)

where $N^2$ is the Brunt Vaisala frequency (BV),

$$N^2 = \frac{-g}{\rho_0} \frac{\partial \sigma_t}{\partial z} \qquad\qquad (8)$$

where g is the gravitational acceleration, $\rho_0$ the average sea water density, z the depth, and $\sigma_t$ is

$\rho$-1000 where $\rho$ is the sea water density. The denominator term $\partial u/\partial z$ in (7) is the velocity gradient, which is an indicator of strength of mechanical generation calculated from vertical current profiles acquired using ADCP.

The low BV (avg. $3.165 \times 10^{-5}$ s$^{-1}$) and large velocity gradient (avg. 3.968 s$^{-2}$) resulted into low Ri (avg. 0.0001), indicating unstable well mixed water column. These lead to instability in the water column and favor eddy-like perturbation in the region.

Instability arises either as a result of mixing of different water masses or due to the shear flows.

Mixing with other water masses can be ruled out as there is clear evidence of the presence of

BoB water in the eddy region from the T-S profiles. Another possibility is the prevalence of planetary waves that might modulate the horizontal flow and induce shear, thereby causing instability; such instability has been well reported along this region by Schott et al., 2009 and

Rao et al., 2010, that planetary waves influence the near surface circulation through local and remote forcing. The role of such planetary wave influence on eddy generation mechanism was examined using altimeter data and mapping of planetary wave propagation was carried out to identify their influence on regional circulation. Referring to Yu (2003), Hovmuller diagram of

SSHA at 8°N along 89°E to 94°E was analyzed to track the planetary wave and are plotted (Fig.

4). Low SSHA in this region from mid-November to mid-January indicates the presence of upwelling mode Rossby wave (Girishkumar et al., 2011). Negative SSHA is almost horizontal, indicating a fast propagation of Rossby wave. Further west (nearer to the eddy location), negative SSHA showed a steeper slope, indicating a slower propagation. The westward propagating signal takes about 45-60 days to travel from the coast of Nicobar Island chain (Potemra et al., 1991) to the core of the eddy region, which yields phase velocity of the westward signal at 0.20 m/s. The theoretical phase speed of Rossby wave at 8°N that propagates westwards is calculated as 0.21 m/s, suggesting that the signal appearing in the plot is a Rossby wave that is generated on the west coast of Nicobar island chain. The estimated speed of the wave is close to the theoretical wave speed and the estimate also compares well with earlier results of Yang et al. (1998), Yu (2003) and Girishkumar et al. (2011). The Rossby waves were produced by radiation from the west coast of Nicobar Island chain in association with poleward
propagating coastal Kelvin waves (Potemra et al., 1991). The baroclinic instability due to the
interaction of westward propagating Rossby waves and local wind stress curl cause meanders
and eddies in BoB (Nuncio and Prasanna Kumar, 2012). Using a numerical model, Kurien et al.
(2010) also concluded that baroclinic instability plays a key role in meander growth and eddy
generation in BoB. Sreenivas et al. (2012) argued that coastal Kelvin waves and the associated
radiated Rossby waves from the east play a dominant role in the mesoscale eddy generation in
BoB. Chen et al. (2012) studied the interannual variability mechanism of the mesoscale eddies
in BoB and pointed that the eddy activities do not directly link to El Nino Southern Oscillation
(ENSO) events and are sensitive to the baroclinic instability of the background flow.

To ascertain the periodicity of SSHA, the data is again subjected to continuous wavelet
transforms with Morlet wave as mother wavelet following Torrence and Compo (1998). It is
clear from Fig. 5 that the dominant mode of variability is semiannual. In the Andaman waters,
the wave period is more variable due to the effect of westward propagating Rossby wave from
the coastally trapped Kelvin wave (Vialard et al., 2009 and Nienhaus et al., 2012). From power
and global wavelet spectrum, the predominant frequencies are in semiannual and annual modes.
The annual mode seems to be reduced in intensity compared to the semiannual mode. On the
basis of the results of wavelet analyses, it is clear that the semiannual Rossby waves are
significant in the years 2005, 2008, 2010 and 2011, whereas the annual wavelets are significant
during 2006-2009. Therefore, we concluded that the westward propagating Rossby wave
radiated from the coastal Kelvin wave contribute to cyclonic eddy in the region.

**Chemical and biological response of the eddy**

Concurrent with the thermohaline oscillations, the vertical structure of dissolved oxygen
(DO) also demonstrates fluctuations above 90 m depth.  The 4.22 ml/L DO contour shoaled
from a depth of about 47 m (92.3°E) to 25-30 m at eastern flank of the eddy (93.3°E). The upper
nitrate ($NO_3$) concentration is in detectable levels (0.67-0.98 µM) and shows slight upsloping
towards the eastern flank (93.3°E). The phosphate ($PO_4$) concentration in the upper water was
also at a detectable level and showed a slight upsloping towards the eastern side (0.12 µM at
92.3°E and 0.27 µM at 93.3°E). Further, the vertical distribution of silicate ($SiO_4$) showed slight
upsloping towards the eastern periphery (0.77 µM at 92.3°E to 1.62 µM at 93.3°E). Hence, concomitant with the thermohaline characteristics, the vertical distribution of nutrients also
showed oscillations in the upper water column.

The physical and chemical characteristics do reflect on the regional biology and this is
well reflected in the surface chl a distribution. Chl a derived from ocean colour imagery (Fig. 6)
can illustrate the standing stock of the primary consumers for the optical depth and is 0.5 mg/m$^3$
in the eddy region compared to the adjacent regions (0.1 mg/m$^3$). This increases within the eddy
in association with the nutrient values explains the impact of churning due to the eddy. And this
points out the significance of such mesoscale processes that influence the production marginally
in the Andaman waters.

**Satellite evidence (SSHA based) for cyclonic eddies**

The distribution of mesoscale production favourable pockets is examined using monthly
SSHA and geostrophic current pattern (Fig. 7a-d) for the winter monsoon (November-February,
2011). This evidences the presence of one cyclonic eddy (CE), of which CE1 is the same that
encountered during the in situ measurements. CE1 was stronger as indicated by negative SSHA
between 5°–9°N with core at 7° N latitude and is observed to be propagating from 93°E to 86°E
within one month (November to December). The eddy intensity is more during November and
December, with a negative value of ⁻0.14m. In December CE1 propagates westward to BoB and
is observed between 86°-93°E.  It is completely replaced from Andaman waters by January and
exhibited a positive SSHA (0.18 m). But the low SSHA observed in BoB waters even during
February centered at 86° E. The shape of the eddy is elliptical with its axis oriented in east west
direction. The eddy CE1 characteristics and generating mechanism is described in the above
sections (3.3.1-3.3.3) using in situ as well as satellite observations.

The SSHA maps also revealed a low SSHA pocket located at 13°N and 93°E during
November with negative anomaly of ⁻0.12m. This is marked as CE2. The negative anomaly is
more in November with SSHA of –0.12 m, and the intensity decreases during December with
SSHA of ⁻0.10. Negative anomaly is replaced by positive anomaly of 0.16m during January.

In order to identify eddies in a prominent way, Okubo-Weiss (OW) parameter method is
also exercised in this study. Eddies are characterized with negative OW parameter at the eddy
core due to the dominance of vorticity over strain components; while strain dominated areas have positive OW parameter. According to Isern-Fontanet et al. (2003), closed contours of OW

with a value of $^{-}2\times10^{-11}$ /s$^2$ corresponding to the threshold value for defining eddies. The threshold value was fixed as same as Isern-Fontanet et al. (2003) for defining eddies and finding out the vorticity dominated area. From the Fig. 3.7, the closed contours of OW, and cyclonic current structure confirmed the presence of an intensified cyclonic eddy at 8°N and 93°E. But the area characterized with threshold value less than $^{-}2\times10^{-11}$ /s$^2$, negative SSHA and the cyclonic current structure at 13°N and 93°E indicated the presence of a weak eddy.

Having recognized eddies from SSHA, OW and geostrophic current maps, it is further confirmed the occurrence of prevailing processes using SST and chlorophyll. Cyclonic eddies formed due to the divergent forcing at the center is occupied with sub-surface nutrient rich waters at the core. These areas of negative SSHA are characterised with relatively cool and high chlorophyll concentration.

SST is high during the initial phase of winter months, i.e. in November (Fig. 8a), with higher values in the entire region of Andaman waters (28.2-28.8 °C). During December (Fig.

8b), the values change to 27.6-28.8 °C. Further, during January (Fig. 8c) and February (Fig. 8d), the basin wide temperature is in the range to 27-29 °C and 26-29 °C respectively.  Though the

Andaman waters are warm in general, the cold core eddies identified in this area show relatively cool temperatures owing to the prevalent cyclonic flow associated with it. CE1 records a temperature of 28.6 °C during November, and when the eddy advances to the Andaman waters the surface temperatures begin to cool. SST decreases from 28.6 to 28.2 °C during December;

SST again decreases to 27.6 °C in January. But in February the temperature remains the same as in January. CE2 displays a temperature of 28.6 °C during November; during December, the temperature decreases to 28.2 °C, and  decreases further to 27 °C during January and again in

February (26.5 °C). The hike in temperature along the eastern Andaman waters might be due to the intrusion of low saline waters through Malacca strait as inferred by Rama Raju et al., 1981, and Tan et al., 2006.

High chlorophyll concentration is expected in the eddy region due to enhancement of nutrients at the surface. These cold core eddies are important because they are in the area of high  biological  activity  and  these  areas  are  observed  to  have  strong  physical  and biogeochemical coupling resulting in high chlorophyll concentration. Generally, Andaman waters are oligotrophic in nature with less chlorophyll concentrations (Vijayalakshmi et al., 2010). The existence of cyclonic circulation increases the chl a levels in the eddy region. When the cyclonic flow advances, increased chl a level was observed in the eddy locations at CE1 and CE2. CE1 recorded 0.1 mg/m$^3$ during November, increased to 0.8 mg/m$^3$ during December and decreased again to 0.3 mg/m$^3$ in January (Fig. 8 a-d). Chl a level decreased to 0.2 mg/m$^3$ in February. CE2 revealed a very low value (0.1 mg/m$^3$) during November; during December, chl a began to increase in the eddy region (0.4 mg/m$^3$) and in January also the pattern followed with a concentration of 0.4 mg/m$^3$, which decreased to 0.2 mg/m$^3$ in February.

The role of wind stress curl on inducing the eddy was verified with weekly progress in the wind stress curl (ASCAT) for the pockets. At CE1 the curl varied from $^-4.43\text{x}10^{-7}$ to $1.28\text{x}10^{-6}$ Pa/m, but the mode of the signal was $^-1.47\text{x}10^{-7}$ Pa/m. The wind curl at CE2 showed values between $^-2.87\text{x}10^{-7}$ and $2.09\text{x}10^{-6}$ Pa/m and mode was $^-3.25\text{x}10^{-8}$ Pa/m. However, the occurrence of maximum negative values implies that wind is not a dominant causative factor for the generation of eddy.

At CE2, the surface temperature is low (27-27.2 °C) compared to the nearby locations and the MLD is also deeper (>70m). Wind is northeasterly, with a magnitude of 4 to 7 m/s. Specific humidity of 14 to 18 g/kg implies dry continental air during the period. Net heat flux varies from $^-98$ to $^-134$ W/m$^2$ during November-February. This causes heat loss due to evaporation (latent heat flux $^-220$ to $^-312$ W/m$^2$), resulting in cooling in the sea surface. Solar radiation varies from 114 to 170 W/m$^2$ in the eddy region. This low solar insolation reduces the SST, resulting densification of water. Thus, the surface water sinks and nutrient rich water entrains from deeper depths. This evidences that the atmospheric forcing causes surface cooling and the resulting convective mixing entrains nutrients into the upper layer which activates the primary production (Prasanna Kumar and Prasad, 1996, Madhupratap et al., 1996). Chatterjee et al. (2016) reported that the equatorial signal of Kelvin comes into the Andaman Sea through the Great Channel, travels along the eastern boundary, and exits to BoB through Preparis Channel, with a smaller part flowing southward along the east coast of the Andaman Islands. In this context it is presumed that the generating mechanism of CE2 is Kelvin. The instability owing to the flow from Ayeyarwady-Salween river system is also supposed to be the reason for CE2

origin.

**Conclusion**

The column dynamics, forcing mechanisms, chemical and biological responses of cyclonic eddies are explained for the Andaman waters based on a suit of in situ and satellite datasets. The eddies are tracked using Okubo-Weiss parameter and the eddy CE1 is strong compared to CE2

based on  the threshold  Okubo-Weiss parameter of $^{-}2x10^{-11}/s^{2}$. The processes are small scale in nature within 100-350 km diameter, and are found to be induced as a result of baroclinic instability arising owing to the westward propagating Rossby wave, semi-annual mode with phase speed of 0.20 m/s for CE1 and CE2 may be induced by Kelvin and the instability occurs due to the Ayeyarwady-Salween flow. While CE2 is associated with the process of convective mixing process occurring in the region due to cold dry continental air from north east. The study concludes that, in addition to the mesoscale processes, the convective mixing occurring along the northwest coast of Andaman is taking a substantial role in triggering the biological production of Andaman waters. Considerable increases in the regional surface biological production indicates the complementary role of such processes in bringing up the quality of production in Andaman waters. The role of convective mixing and eddies in the dynamics of the

Andaman waters are explained for the first time.

**Acknowledgements**

Authors    are    grateful    to    the    Ministry    of    Earth    Sciences    for    supporting the    work    and    for    providing    facilities    onboard    FORV    Sagar    Sampada    for    in-situ measurements.    All    the    fellow    participants    of    the    cruise    FORV    SS292    are thankfully acknowledged.  Insitu data are obtained from FORV Data Centre in CMLRE.ASCAT

Scatterometer wind field is obtained from NOAA/NESDIS.  The TOPEX/Poseidon SSHA

product is generated from the Merged Geophysical Data Record.  Chlorophyll data was retrieved from GSFC NASA. Heat flux data is provided by WHOI OAFlux project.

[Figure]

Fig. 1 Station Location

[Figure]

Fig. 2 a) Sea Surface Height (cm- Aviso weekly) and geostrophic current (cm/s) and the eddy location b) Vertical temperature (°C), c) salinity and d) density (kg/m$^3$) distribution at the eddy location

[Figure]

Fig. 3  Horizontal current (m/s) structure at different depths along 8°N

[Figure]

Fig. 4  Hovmuller diagram of SSHA(m) (Aviso monthly) along 8°N

[Figure]

Fig. 5. Wavelet spectra of SSHA (m- Aviso monthly from 2003-2013) along 8°N

[Figure]

Fig. 6 chl a (mg/m$^3$- weekly MODIS Aqua) pattern during the insitu observation

[Figure]

Fig. 7 Merged map of  SSHA (m), Geostrophic current (cm/s) and Okubo-Weiss paremeter (Black contour of -2x10-11/s2) from Aviso during a) November b) December c) January d) February

[Figure]

Fig.8 Overlap map of SST (°C-monthly MODIS Aqua) and Chl a (mg/m³-monthly MODIS Aqua) during a) November, b) December, c) January, d) February

---

## Author Comment (AC2) · 13 Dec 2018

Reviewer 2 1. Observation-based analysis (Figs 2 and 3): I'm not convinced you sampled an eddy in the stations you show. Yes, there is some northward flow to the west and some southward flow to the east, but it is hard to confirm it is an eddy on not only a current interacting with the continental shelf. The SSH map cannot confirm this is an eddy - it does not resolve this scale (only features that are larger than 110 km in radius) We changed the SSHA-geostrophic current figure and adopted Okubo-Weiss method to track the eddy. From this we confirmed the presence of cyclonic eddy.

2. Even if you still think that the ADCP data and the SSH maps indicate the presence

of a sub-mesoscale eddy, this eddy is anticyclonic (clockwise rotation in the northern hemisphere). Therefore, all the discussion and data interpretation that points this eddy as being cyclonic is erroneous (e.g., lines 144, 152). Please, re-interpret your data keeping this in mind.

SSHA and geostrophic current plot confirmed the presence of cyclonic eddy in the region

3. I could not understand why the authors show the vertical sections of T, S, and density only down to 200 m in Figure 2. If these variables where sampled by the CTD, I would expect deeper measurements. If you look at values below 200 m, you might get more insight about the structure you sampled.

Included the same in the revised manuscript

4. Still about Figure 2: the max and min values in the colour axes in (b), (c), and (d) are not appropriate. This choice might be hindering some isolines above 40 m. Please review this figure. Reviewed the figure as per the suggestion of the reviewer

5.I could not understand the advances this manuscript brings to the Rossby Wave propagation and eddy triggering to the literature. Please state it in the manuscript. 6. I was not convinced that the Rossby Waves indeed triggered the eddy. You need more results to support this claim. The whole section on "Generation Eddy Mechanism" needs through review and more results to confirm your claims. The statement in lines 218-220 needs proof to be accepted.

Answer to 5 and 6: The hovmuller diagram on SSHA zonally along 8ïĆřN indicate the westward propagation of Rossby crossing the eddy area which modify the regional dynamics. Forcing mechanism of the eddy is identified as due to the baroclinic instability due to vertical shear in the horizontal flow; one of the reason for which is the westward propagating Rossby wave (Nuncio and Prasannakumar, 2012)

7. Figure 6 shows higher [chla] close to the islands. You claim this is because of eddy

effect. It just looks like it is a natural coastal increase in nutrients (river runoff, upwelling, current-bottom friction). Is this region of the world different, and these processes would not be in place?

This high chla near island chain is due to the island effect and extened high patch towards offshore is due to the strong cyclonic eddy as indicated by the high Okubo-Weiss parameter.

8. Figure 8 actually suggests an increase in [chla] caused by the presence of an eddy. See the spiralling green patch between 11-14N and 85-88E. This might relate to a cyclonic eddy.

This is a cyclonic eddy present in BoB during winter which already discussed in detail by Vinayachandran and Simi (2003)

9. The part of the manuscript that requires a specially careful analysis and interpretation is the SSHA analysis (Figure 7). The SSH product used does not resolve the features you are trying to investigate. You need to zoom out to look at the mesoscale eddies. In addition, an eddy is defined by a closed SSH contour. You cannot see this in any of the features you indicate as "eddies". All the paragraphs in the manuscript related to this figure must be intensively corrected.

Modified the plot on SSHA and geostrophic current and included in the revised manuscript

10. I could not comment on the biogeochemical results (Table 1) and in the wind stress results because they are not presented in a suitable manner. Please make a figure with the values in Table 1 and a figure with the wind results if included in the next manuscript.

11. The domains you look at in Figures 1, 2, 6, 7, and 8 are all different in space - and probably in time (but I can't tell because this information is not given). You cannot discuss the "eddies" from these different datasets as you do here because they are

not the same ones! In order to explain the basin scale processes and link this into the insitu observations.

Minor comments

7. Lines 50-52: does this relate to Andaman eddies or eddies in general? Explained in the context of BoB but the case is general

8. Lines 61-63: does this relate to Burnaprathepart 2010 eddy or to the eddy you describe in this manuscript?

It is the eddy described by Burnaprathepart 2010

9.In the Data and Methods section, you should only include what you used in your manuscript. For example, you did not analyse AVISO data between 2003 to 2013.

The data used for wavelet analysis 11. Please describe in the manuscript the reason for working with the weekly AVISO dataset, instead of the daily product. We have taken weekly product to avoid small scale variation 12. Line 107: Before describing the wavelet analysis, it helps the reader if you write a brief line saying what you use it for later on the manuscript. In the present study the wavelet is applied to explain the temporal variation of SSHA in the eddy region to explore the life span and frequency of the processes during the 10 years. 15. Line 151: The figure does not show this is a sub-surface cyclonic eddy. You are not resolving this feature neither in the horizontal direction or in the vertical direction. The temperature profile indicates the core of the eddy in the subsurface (40-60 m) along the transect. The ADCP derived current gives the measurements from 16-88 m only. However the SSHA derived geostrophic current indicate the presence of eddy in the surface water also. These gives the indication that the core lies in the sub-surface with its influence extending tto surface.

Please also note the supplement to this comment:
https://www.ocean-sci-discuss.net/os-2018-23/os-2018-23-AC2-supplement.pdf

[Figure]

[Figure]

Fig. 1 Station Location

**Fig. 1.** Fig. 1 Station Location

[Figure]

Fig. 2 a) Sea Surface Height (cm- Aviso weekly) and geostrophic current (cm/s) and the eddy location b) Vertical temperature (°C), c) salinity and d) density (kg/m³) distribution at the eddy location

**Fig. 2.** Fig. 2 a) Sea Surface Height (cm- Aviso weekly) and geostrophic current (cm/s) and the eddy location b) Vertical temperature (ïĆřC), c) salinity and d) density (kg/m3) distribution at the

[Figure]

Fig. 3   Horizontal current (m/s) structure at different depth at 8°N

**Fig. 3.** Fig. 3 Horizontal current (m/s) structure at different depths along 8°N

[Figure]

Fig. 4  Hovmuller diagram of SSHA(m) (Aviso monthly) along 8°N

**Fig. 4.** Fig. 4 Hovmuller diagram of SSHA(m) (Aviso monthly) along 8°N

[Figure]

Fig. 5. Wavelet spectra of SSHA (m- Aviso monthly from 2003-2013) along 8°N

**Fig. 5.** Fig. 5. Wavelet spectra of SSHA (m- Aviso monthly from 2003-2013) along 8°N

[Figure]

Fig. 6 chl a (mg/m³- weekly MODIS Aqua) pattern during the insitu
observation

**Fig. 6.** Fig. 6 chl a (mg/m3- weekly MODIS Aqua) pattern during the insitu observation

[Figure]

Fig. 7 Merged map of  SSHA (m), Geostrophic current (cm/s) and Okubo-Weiss paremeter (Black contour of -2x10-11/s2) from Aviso during a) November b) December c) January d) February

**Fig. 7.** Fig. 7 Merged map of SSHA (m), Geostrophic current (cm/s) and Okubo-Weiss pareme-ter (Black contour of -2x10-11/s2) from Aviso during a) November b) December c) January d) February

[Figure]

Fig.8  Overlap map of SST (°C-monthly MODIS Aqua) and Chl a (mg/m³- monthly
MODIS Aqua) during a) November, b) December, c) January, d)  February

**Fig. 8.** Fig.8 Overlap map of SST (ïĆřC-monthly MODIS Aqua) and Chl a (mg/m3- monthly
MODIS Aqua) during a) November, b) December, c) January, d) February